# LiteQUIC: Improving QoE of Video Streams by Reducing CPU Overhead of QUIC

Pengqiang Bi
Shandong University
Qingdao, China
pq.bi@mail.sdu.edu.cn

Yifei Zou*
Shandong University
Qingdao, China
yfzou@sdu.edu.cn

Mengbai Xiao*
Shandong University
Qingdao, China
xiaomb@sdu.edu.cn

Dongxiao Yu
Shandong University
Qingdao, China
dxyu@sdu.edu.cn

Yijun Li
Baishan Cloud
Guiyang, China
yijun.li@baishan.com

Zhixiong Liu
Baishan Cloud
Guiyang, China
zhixiong.liu@baishan.com

Qun Xie
Baishan Cloud
Guiyang, China
qun.xie@baishan.com

## Abstract

QUIC is the underlying protocol of the next generation HTTP/3, serving as the major vehicle delivering video data nowadays. As a userspace protocol based on UDP, QUIC features low transmission latency and has been widely deployed by content providers. However, the high computational overhead of QUIC shifts system knobs to CPUs in high-bandwidth scenarios. When CPU resources become the constraint, HTTP/3 exhibits even lower throughput than HTTP/1.1. In this paper, we carefully analyze the performance bottleneck of QUIC and find it results from ACK processing, packet sending, and data encryption. By reducing the ACK frequency, activating UDP generic segmentation offload (GSO), and incorporating PicoTLS, a high-performance encryption library, the CPU overhead of QUIC could be effectively reduced in stable network environments. However, simply reducing the ACK frequency also impairs the transmission throughput of QUIC under poor network conditions. To solve this, we develop LiteQUIC, which involves two mechanisms towards alleviating the overhead of ACK processing in addition to GSO and PicoTLS. We evaluate LiteQUIC in the DASH-based video streaming, and the results show that LiteQUIC achieves 1.2× higher average bitrate and 93.3% lower rebuffering time than an optimized version of QUIC with GSO and PicoTLS.

## CCS Concepts

• **Networks → Transport protocols**; **Network performance analysis**; • **Information systems → Multimedia streaming**.

## Keywords

HTTP/3, CPU overhead, DASH, ACK, GSO

---

*Corresponding author.

**ACM Reference Format:**
Pengqiang Bi, Yifei Zou, Mengbai Xiao, Dongxiao Yu, Yijun Li, Zhixiong Liu, and Qun Xie. 2024. LiteQUIC: Improving QoE of Video Streams by Reducing CPU Overhead of QUIC. In *Proceedings of the 32nd ACM International Conference on Multimedia (MM '24), October 28-November 1, 2024, Melbourne, VIC, Australia.* ACM, New York, NY, USA, 10 pages. https://doi.org/10.1145/3664647.3681670

## 1 Introduction

Video traffic has accounted for 65.9% of the Internet traffic as of 2022 [57], where HTTP-based adaptive bitrate (ABR) streaming is the de facto standard [60]. As the underlying protocol of HTTP/3 [8], QUIC [39] has been widely deployed by content providers. In Facebook, QUIC delivers 75% of its traffic and higher quality of user experience (QoE) is received [57]. YouTube stands out as the largest user of QUIC, experiencing a reduction of over 9% in rebuffering time and an increase of over 3% in throughput [19].

Despite the superior performance over TCP observed under poor network conditions [55, 56, 58, 64, 67], QUIC is noticed to have inferior transmission throughput in high-speed networks due to its high computational cost [34, 38, 68]. We try exposing this problem with a preliminary experiment, where we run DASH-based streaming sessions using HTTP/1.1 based on TCP and HTTP/3 based on QUIC. We measure the average bitrate and rebuffering time of streaming sessions, and present the results in Figure 1. With increasing clients connected to the server, HTTP/3 sessions experience degrading throughput and increasing rebuffering time while HTTP/1.1 ones do not. When there are 100 clients simultaneously fetching video data from the server, HTTP/3 has only 40% average bitrate and 21.5× more rebuffering time compared with HTTP/1.1. It is worth noting that in the experiment, the available bandwidth is always greater than the demands of all clients so that the QoE degradation is attributed to the excessive CPU overhead of QUIC.

The computational cost of QUIC can be mainly categorized into three parts: **(1) ACK processing:** To ensure reliable transmission, QUIC clients send acknowledgments (ACKs) upon receiving packets. However, as a userspace transport protocol, QUIC needs to frequently read ACKs through system calls, which introduces non-negligible overhead. Moreover, as QUIC packets are small (∼1KB), processing packet-wise ACKs also demands a large proportion of the computational cost. ACK processing plays a major role in incurring excessive CPU overhead compared with TCP. **(2) packet**

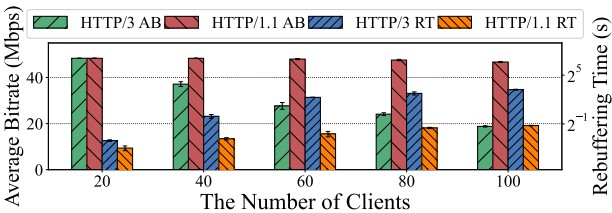

**Figure 1:** The average bitrate (AB) and rebuffering time (RT) of DASH-based streaming sessions using HTTP/3 and HTTP/1.1. The experimental setup is described in Section 5.2.

**sending:** Unlike TCP that does not limit the data length written through a system call and assemble packets in the kernel, QUIC has to organize the small QUIC packets in the userspace and sends one each system call. The frequent system calls result in significant overhead as well. **(3) data encryption:** In HTTP/1.1, data encryption occurs at the TLS layer above TCP, allowing to encrypt a large bulk of payload (up to 64KB) at a time. On the other hand, QUIC integrates the encryption module itself, and it encrypts packets individually. Switching from bulk-based encryption to packet-based means more function calls thus higher CPU overhead. Additionally, QUIC encrypts packet headers while TLS does not, which also introduces additional overhead.

In this paper, we first attempt to reduce the high CPU overhead of QUIC by reducing ACK frequency, activating UDP generic segmentation offload, and incorporating a high-performance encryption library, PicoTLS [25]. we find that simply reducing ACK frequency can impair the transmission performance in poor network conditions because the long queue at the client throttles the in-flight data. This also requires modifications to the client, which is often not feasible as most clients are browsers. Therefore, in addition to GSO and PicoTLS, we design a server-side ACK merging mechanism and an adaptive mechanism of adjusting ACK frequency at the client-side. In the CPU-limited state, LiteQUIC reads multiple ACKs that have arrived in a batch from the kernel space and merges them for processing. The ACK merge mechanism could also benefit GSO, which does not work when the ACK frequency is high. Additionally, if the client is allowed to install LiteQUIC, the ACK frequency is adaptively adjusted to further reduce the overhead and processing ACKs without degrading transmission throughput. LiteQUIC achieves up to 1.78× higher transmission throughput than picoquic [51], a QUIC implementation featuring high-performance, in the CPU-limited scenarios, and the throughput of LiteQUIC does not decrease in weak network environments as vanilla HTTP/3. When deployed in DASH-based video streaming sessions, LiteQUIC improves the average bitrate by 20.5% and reduces the rebuffering time by 93.3% compared with an optimized version of QUIC with GSO and PicoTLS. The contributions of this paper are as follows:

- By carefully analyzing QUIC, we discover that its excessive computational cost results from ACK processing, packet sending, and data encryption, and simply reducing ACK frequency impairs the transmission performance in poor networks.
- We design and implement LiteQUIC, which involves two ACK processing mechanisms towards alleviating the overhead of ACK processing without impairing transmission performance.

- We evaluate LiteQUIC in both data transmission and DASH-based streaming sessions, and the results show that LiteQUIC outperforms the state-of-the-art QUIC implementations, and it improves the QoE as a large number of clients are connected.

In Section 2, We discuss related work, and in Section 3, we analyze the CPU overhead as well as the impact caused by ACK frequency in HTTP/3. The design of LiteQUIC is presented in Section 4, along with its evaluation in Section 5. Finally, Section 6 concludes our work.

## 2 Related Work

### 2.1 High Performance Network Stack

Nowadays, the Linux network stack are struggling to keep up with rapidly growing link bandwidth, and the CPU resources become the bottleneck of data transmission. There are a number of studies devoted to solving this problem, including network stack optimization [27, 30, 41], hardware offloading techniques [3, 23], and kernel-bypass approaches [46, 66]. There are also optimization schemes for different application scenarios, such as SPRIGHT [52] proposed for improving serverless computing and dcPIM [9] for data transmission in datacenters. A recent work of Cai et al. [10] analyzes the overhead of kernel stacks and reports that TCP can achieve ~42Gbps throughput by leveraging techniques such as segmentation and receive offload, jumbo frames, and packet steering with commodity NICs. In their follow-up work [11], they propose NetChannel, which leverages the modern multicore architecture to saturate link bandwidth more than 100Gbps over a single socket.

### 2.2 QUIC Protocol

Recently, Yang et al. [65], compare four QUIC implementations for dissecting their CPU overhead, and propose offloading operations such as partial encryption and packet reordering to NICs to reduce the overhead. Other researches [29, 31, 50] also explore offloading TLS encryption to hardware. The performance issue of QUIC caused by ACKs has been studied [16, 17, 37, 43, 44], where Marx et al. [44] demonstrate that most implementations currently send an ACK for every 2-10 received packets instead of the recommended 2, and other studies [16, 17, 37] show that the ACK threshold of 2 imposes significant CPU overhead on the sender, while setting it to 10 can alleviate this issue. Liu et al. [43] design a strategy of dynamically adjusting ACK frequency based on Bandwidth-Delay-Product (BDP) to reduce the overhead. In addition to the aforementioned works, we not only show the impact of different ACK frequencies in various network environments, but also analyze the reasons for these differences. Moreover, the server-side ACK merging approach we designed does not require client-side support.

### 2.3 Video Streaning over QUIC

Since QUIC is proposed, a large body of studies explore if it could benefit video streaming. A study [6] shows that QUIC-based DASH video streaming does not necessarily result in QoE enhancement, while another thread of studies [4, 14, 15] demonstrate that QUIC outperforms TCP in lossy networks. This is because QUIC enables packet multiplexing that removes head of line blocking and it has up

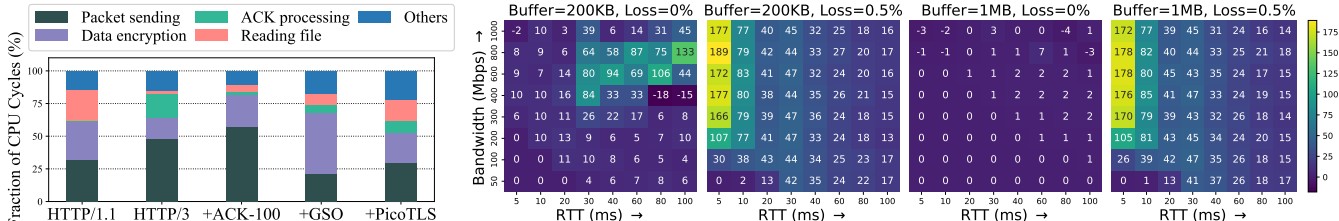

**Figure 2: Left:** The CPU overhead of different components in HTTP/1.1, vanilla HTTP/3, and HTTP/3 with optimization techniques. +ACK-100 sets the ACK threshold to 100, +GSO activates GSO on top of +ACK-100, and +PicoTLS incorporates the high-performance PicoTLS. **Right:** The throughput gain of setting ACK threshold to 2 over setting to 100.

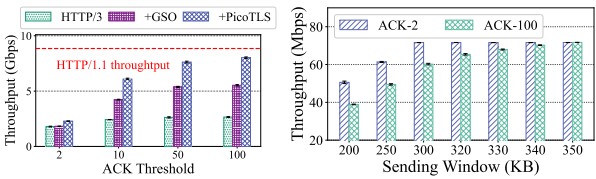

**Figure 3: Left:** The throughput of HTTP/3 with different ACK thresholds. The red line indicates the throughput of HTTP/1.1. **Right:** The impact of varying size of the sending window on transmission throughput when the network BDP is fixed to 300KB.

to 256 NACKs resulting in fast loss recovery, both alleviating the impact of poor network conditions on video playback [5]. Shreedhar et al. [59] observe that QUIC is faster in establishing connections with video servers compared to TCP+TLS and reduces stall durations, especially in lossy networks. And another study [63] evaluates when multiple connections exist, QUIC reaches higher QoE scores than TCP as the number of connections increases. In our work, multiple connected clients are considered one of reasons imposing excessive computational overhead to the server, which then throttles the network throughput and impairs QoE. We aim to reduce the CPU overhead of QUIC to mitigate this.

## 3 HTTP/3 Analysis and Optimization

In this section, we test transmission sessions using HTTP/3 for analyzing its CPU overhead and identifying performance bottlenecks. **Testbed:** We set up the testbed using two machines directly connected via a 10Gbps link. Both machines are equipped with 10Gbps LREC6880BT NICs and run Ubuntu 20.04 LTS with Linux kernel 5.15.0. We evaluate the performance in the client-server mode. The server uses the nginx-quic [48], which supports both HTTP/1.1 and HTTP/3, and it has an Intel Xeon CPU at 2.90GHz. The client employs nghttp2 [47], a stress testing tool that can evaluate both HTTP/1.1 and HTTP/3, running in a docker environment on the other machine with an Intel Xeon CPU at 2.10GHz. **Experimental metrics:** We launch ten clients concurrently to request a 1GB file and analyze the CPU overheads using perf [21]. We measure the network throughput and CPU overhead of HTTP/1.1 and HTTP/3 sessions. The CPU overhead of protocols are summarized as follows: **1) Packet sending:** This is the overhead of sending TCP packets in HTTP/1.1 and sending UDP packets in

HTTP/3. For TCP, we measure `sock_write()`, which eventually calls `tcp_sendmsg()`. For QUIC, we measure `ngx_quic_send()`, which eventually calls `udp_sendmsg()`.**2) Data encryption:** This is mainly incurred by encryption modules in the protocol stack. For HTTP/1.1, we measure `tls_seal_record()`. For HTTP/3, we measure `ngx_quic_encrypt()`, which contains `ngx_quic_tls_seal()` and `ngx_quic_tls_hp()` to encrypt the payloads and headers of the QUIC packets, respectively.**3) ACK processing:** This involves the overhead of reading ACK frames from the socket, as well as parsing and processing them. For TCP, we measure `tcp_ack()`. For QUIC, we measure `ngx_quic_recvmsg()` that has `recvmsg()` and `ngx_quic_input_handler()`.**4) Reading file:** This is the overhead of reading files from the disk. We measure `ngx_read_file()` for both HTTP/1.1 and HTTP/3.**5) Others:** Any other overhead.

After the transmission experiments, we find that the average throughput of HTTP/3 is 1.79Gbps, which is only 20% of HTTP/1.1. The dissection of CPU overhead of two protocols is presented as the leftmost two pillars in Figure 2. For HTTP/3, its CPU overhead is mainly composed of packet sending, ACK processing and data encryption. The three modules account for 82.83% of total CPU cycles, where packet sending takes 48.24%, ACK processing takes 18.43%, and data encryption takes 16.16%. Compared to HTTP/1.1, ACK processing is the most increased component.

Towards the three major bottlenecks, we first attempt to reduce the CPU overhead by reducing the ACK frequency, activating GSO and incorporating the efficient encryption library PicoTLS.

## 3.1 Reducing ACK Frequency

HTTP/3 incurs higher CPU overhead of ACK processing because ACKs now are processed in the userspace instead of the kernel space as HTTP/1.1. To process ACKs, QUIC has to frequently read them from the kernel to the application layer. *ACK threshold* is defined in RFC 9000 [33] to alleviate such overhead. The client could determine the frequency of acknowledging QUIC packets with ACK threshold. For example, the default value of 2 in RFC means we send an ACK upon receiving two QUIC packets. We measure how ACK frequency impacts the transmission throughput, and the results are shown in the left of Figure 3. Note that the larger the ACK threshold value, the lower the frequency of ACKs sent by the client. When it is set to 100, we find that the throughput of HTTP/3 is 2.65Gbps, which is 1.48× higher than that setting ACK threshold to 2. We also plot CPU overhead of this setting in Figure 2, where we can find that the cycles required by ACK processing is

reduced to 2.65%, and it means that the throughput improvement results from the reduced CPU overhead.

Reducing the frequency of ACKs decreases the CPU overhead at the server, thus improving throughput when CPU resources are constrained. **However, when CPU resources are abundant and bandwidth resources are limited, what are the impacts of reducing ACK frequency?** To answer this, we use $tc$ [2] to set latency from 5ms to 100ms and bandwidth from 50Mbps to 1000Mbps based on values commonly discovered in modern networks [12, 28]. Additionally, we set a 200KB shallow buffer and a 1 MB deep buffer to simulate the bottleneck buffer [7]. To test performance in poor network conditions, we also introduced an additional 0.5% random packet loss ratio. In each scenario, we request a file with a size of 1GB and define $TPGain$ as the throughput gain by comparing the throughput of setting ACK threshold as 2 (ACK-2) to that of setting ACK threshold to 100 (ACK-100):

$$TPGain = \frac{TP_{ACK-2} - TP_{ACK-100}}{TP_{ACK-100}}.$$

We draw the heatmaps of $TPGain$ in different network conditions in the right of Figure 2. We notice that with a shallow bottleneck buffer of 200KB, the throughput of ACK-2 surpasses that of ACK-100 in high-latency and high-bandwidth network conditions. The largest gap occurs at (800Mbps, 100ms), where the throughput of ACK-2 is 1.33× higher than ACK-100. However, in deep bottleneck buffer scenarios, there is little difference between ACK-2 and ACK-100. In networks with random packet loss, the overall throughput of ACK-2 is superior to ACK-100. In the network of (1000Mbps, 5ms), ACK-2 achieves 1.7× higher throughput than ACK-100.

We now analyze these phenomenon. First, when the in-flight data matches BDP of the network, the bandwidth is fully utilized [13]. Additionally, when it surpasses the BDP, the surplus data is queued in the buffer of the bottleneck link. Conversely, when the in-flight data is lower than the BDP, the link bandwidth is underutilized. By setting the ACK threshold to 100, the client delays sending an ACK until 100 packets are received. As a result, the actual in-flight data within the network is

$$inflight = swnd - data_c,$$

where $swnd$ represents the size of sending window at server and $data_c$ denotes the amount of data awaiting at the client. Since $swnd$ is usually converges to a fixed value in a session, $inflight$ is then determined by $data_c$. As more data are queued at the client, the less in-flight data in the network. This can be verified by varying the sending window size. We configure the network with a BDP of 300KB (80Mbps, 30ms), and changes $swnd$ from 200KB to 350KB. We measure the throughput of ACK-2 and ACK-100 and present the results in the right of Figure 3. When the sending window is less than the BDP of 300KB, both configurations fail to fully utilize the bandwidth, and ACK-100 with lower $inflight$ has lower throughput. ACK-2 can fully utilize the bandwidth when the sending window reaches the BDP of 300KB, while ACK-100 requires additional 50KB to cover the data queued at the client. We now interpret the results of the Figure 2. In scenarios with shallow bottleneck buffers, where congestion-induced packet loss leads to a sending window smaller than the BDP, ACK-2 outperforms than ACK-100 because the difference of in-flight data volume. The same explanation applies to

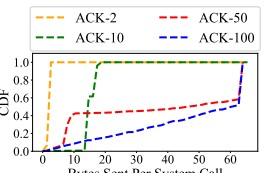 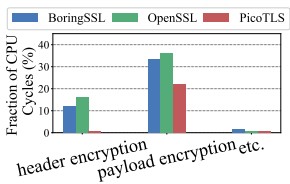

**Figure 4: Left:** The CDF of how many packets are sent per system call with GSO turned on. **Right:** The CPU overhead of components of BoringSSL, OpenSSL and PicoTLS.

scenarios involving random packet loss. In cases with deep buffering, the sending window is approximately the BDP plus the buffer size. Even after accounting for data queued at the client side, the $inflight$ still exceeds the BDP. Consequently, ACK-2 and ACK-100 exhibit the similar performance.

In summary, **while reducing the frequency of ACKs can decrease the overhead of ACK processing, it also leads to inferior transmission throughput in poor network conditions** Additionally, it requires modifications to the client, making it unsuitable for clients those are not open to setup like browsers.

### 3.2 UDP Segmentation Offload

In Figure 2, setting ACK threshold to 100 shifts the bottleneck of HTTP/3 to packet sending, with the overhead of 57.64% compared to 32.03% of HTTP/1.1. Based on TCP, HTTP/1.1 is allowed to send millions of bytes of data to the socket in a single system call, whereas the QUIC protocol used by HTTP/3 is realized to send one QUIC packet (~1KB) per system call. Such frequent system calls consume more CPU cycles.

We attempt to mitigate the overhead of packet sending by leveraging UDP GSO [20], which can combine up to 64 packets in a single system call. **However, we notice that GSO is not effective when the ACK threshold is low.** During a transmission session, the server moves forward its sending window according to the data acknowledged. As a result, When the ACK threshold is low, the sending window is updated in a small but frequent manner. As a result, every time QUIC sends packets to the kernel, the data allowed to be sent are smaller than the GSO capacity, thus failing to reduce the number of system calls. As the ACK threshold increases, more space is released in the sending window after receiving an ACK, and GSO can starts to show its power. In the left of Figure 3, the transmission throughput is barely improved by activating GSO when the ACK threshold is 2. By increasing ACK threshold to 10, 50, and 100, enabling GSO increases the throughput by 0.74×, 1.04×, and 1.09×, respectively. We also collect the number of bytes sent per system call at different ACK thresholds in Figure 4. When the ACK threshold is 10, over 99.4% of system calls can send more than 10KB of data. As the ACK threshold increases to 100, over 50.2% of system calls send data more than 60KB.

### 3.3 Efficient Encryption Library: PicoTLS

By setting ACK threshold to 100 and enabling GSO, the throughput of HTTP/3 reaches 62.7% of HTTP/1.1. The component consuming the most CPU cycles is now data encryption. Encrypting the same length of data in HTTP/3 imposes the CPU overhead of 46.91%

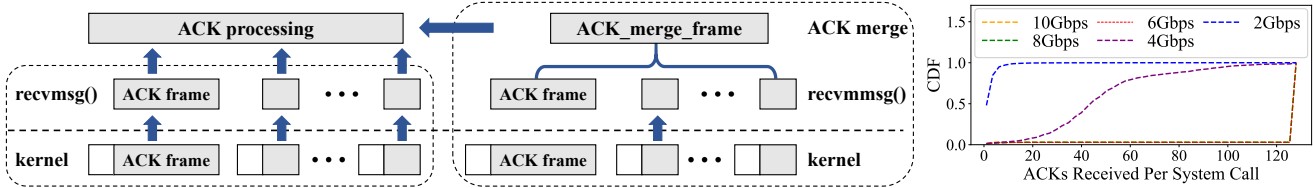

**Figure 5: Left:** How ACK is processed in QUIC and LiteQUIC. QUIC reads and processes an ACK frame a time, and LiteQUIC reads ACK frames in a batch and merge them into one before processing. **Right:** The number of ACKs received per `recvmmsg()` system call. 10Gbps, 8Gbps and 6Gbps correspond to CPU-limited states, while 4Gbps and 2Gbps correspond to bandwidth-limited states.

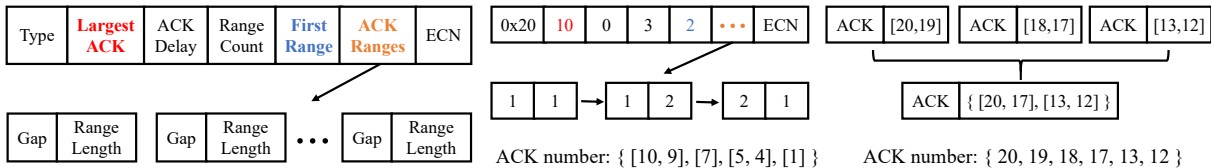

**Figure 6: The format of QUIC ACK frame.** The left is the format, and the middle is an example, which acknowledges packets starting from 10. The right is an example of merging three ACK frames. Before merging, each ACK frame contains an ACK range, and after merging, the only ACK frame contains two ACK ranges.

compared to 28.80% in HTTP/1.1. This discrepancy arises because HTTP/1.1 encrypts data at the SSL layer, allowing the encryption of up to 64KB of data in one pass. In contrast, QUIC encrypts each packet individually, which means more frequent function calls. It is reported that the performance of OpenSSL degrades by around 30% when encrypting small blocks [36]. Furthermore, QUIC encrypts the header of packets, incurring additional overhead.

PicoTLS is a lightweight library realizing TLS protocol. It eliminates the overhead of encrypting QUIC packet headers and reduces the overhead of encrypting small blocks of data from ~30% to ~10% [36]. We integrate PicoTLS into LiteQUIC and compare its performance with BoringSSL and OpenSSL, two widely used encryption libraries. In Figure 4, we investigate the fraction of CPU cycles of different encryption components. The results show that PicoTLS reduces the overhead of encrypting headers from 12.12% to 0.61% and payload encryption overhead from 33.25% to 21.91% compared to BoringSSL. With PicoTLS, the throughput of HTTP/3 reached 8.02Gbps, which is 90.6% of HTTP/1.1 as shown in Figure 3.

## 4 LiteQUIC Design

Reducing ACK frequency could reduce the CPU overhead of QUIC, but it fails to adapt to poor network conditions as discussed in Section 3.1. Furthermore, it requires modifications to the client, which is often not feasible. To address these, we design the following two mechanisms to reduce the overhead of ACK processing:

- *ACK merge*: We read ACK frames in batches and merge them into one for processing at server.
- *Adaptive ACK frequency*: If the client is configurable, we dynamically adjust the ACK frequency, only reducing it in CPU-limited scenarios.

With these two mechanisms and by additionally incorporating GSO and PicoTLS, we have LiteQUIC, a lightweight QUIC that works well in various environments.

### 4.1 ACK Merge

In QUIC, a UDP packet carrying an ACK frame is read from the kernel by `recvmsg()`, which is further parsed for data acknowledgment. However, in high-speed network scenarios, frequent system calls may saturate the CPUs on server. As the CPUs become the bottleneck, a backlog of UDP packets carrying ACKs appears in the server kernel. This could be alleviated by `recvmmsg()` provided by Linux kernel, which can read multiple packets in a single call. With these packets, we then merge the ACK frames parsed from them into one for further processing. The original ACK processing pipeline and the new one are presented in the left of Figure 5.

To merge the ACK frames, we need first understand their format as shown in Figure 6. The fields required in the merge are as follows: 1) **Largest ACK**: The largest packet identifier (starting from 0 in a session) to be acknowledged 2) **First ACK Range**: The number of packets to be acknowledged following the first packet (including the first packet and with decreasing identifier) 3) **ACK Range Count**: The number of acknowledgment ranges. 4) **ACK Range**: It points to the list of acknowledged ranges, Gap means the distance (calculated in packet number) from the previous acknowledgment range, Range Length means the number of packets in the current range. An ACK frame contains multiple acknowledgment ranges. In QUIC, the acknowledgment ranges are processed individually, and by processing a range, the server has to traverse the sending queue for removing packets that have been received. This means even we can retrieve multiple packets using `recvmmsg()`, it still demands excessive CPU cycles to processing ACK range by repeatedly traversing the sending queue. As a result, we merge their ACK ranges before the processing. We give an example of merging ACK ranges in the right of Figure 6, three ACK frames, each with one ACK range, are merged into one frame containing two ranges. With this method, we can quickly clear the backlogged ACK frames in a CPU-constraint scenario.

We test ACK merge by limiting bandwidth with $tc$ to 10Gbps, 8Gbps, 6Gbps, 4Gbps, and 2Gbps. We collect the number of ACK frames received by each `recvmmsg()` and present the results in the right of Figure 5. When the bandwidth is limited to 2Gbps, the CPU resources are abundant, and all arriving ACK frames can be handled promptly. In this case, 48% of `recvmmsg()` read only one ACK frame, and 84% read three or fewer ACK frames. If the bandwidth increases to 4Gbps, The CPU resources do not throttle the transmission throughput only if ACK merge is used (without ACK merge, only 2.28Gbps is achieved as shown in Figure 3) In this case, 80% of `recvmmsg()` read 60 ACK frames or fewer. As the bandwidth exceeds 6Gbps, the server becomes CPU-limited, and more than 97% of `recvmmsg()` read 128 ACK frames, which is the maximum set to read each time. We can see that our ACK merge mechanism effectively reduces the CPU consumption of server and thus improves the transmission throughput.

Additionally, as mentioned in Section 3.1, the default ACK threshold of 2 fails the GSO optimization because it pushes forward the sending window in a frequent but small manner. However, with this new mechanism, processing the merged ACK frames allows the sending window to release a wide range of space, enabling GSO to combine enough packets.

## 4.2 Adaptive ACK Frequency

The server-side ACK merge mechanism does not require any modifications to the client and can be easily deployed. But if the client is allowed to be changed, we design an adaptive ACK frequency mechanism to further reduce the CPU overhead of server. This mechanism dynamically adjusts the ACK frequency based on network conditions, **only reducing it when the server CPU resources are constrained**. We utilize the extension specified in a working IETF draft [32] to dynamically adjust the ACK frequency in a session, which designs a `ACK-FREQUENCY` frame. At the beginning of a session, the server and client exchange the newly defined transport parameter, min_ACK-delay to notify each other of their support to this extension. If both parties support it, the server can send a `ACK-FREQUENCY` frame to the client at any time, informing it to modify the ACK threshold. When the server is in the CPU-limited state, most `recvmmsg()` read multiple ACK frames. Therefore, we calculate the average number of ACK frames read by `recvmmsg()` per RTT as $ACK_{avg}$ and use it to determine that

$$state = \begin{cases} CPU\text{-}limited, & if\ ACK_{avg} \geq 2 \\ bandwidth\text{-}limited, & if\ ACK_{avg} < 2 \end{cases}$$

When $ACK_{avg}$ exceeds 2, it indicates that the processing of ACK frames on the server is slower than the arrival of ACKs. We consider this to be a CPU-limited state, and at this point, we adjust the ACK threshold on the client to

$$T_{new} = \lfloor T_{old} \times ACK_{avg} \rfloor,$$

where $T_{new}$ is the updated ACK threshold and $T_{old}$ is the old value. This implies that the client should reduce the ACK frequency because the server expects to receive one ACK per system call, which consumes the least CPU resources. When $ACK_{avg}$ is less than 2, we consider it to be in a bandwidth-limited state, and we update the

ACK threshold to

$$T_{new} = \max(\lfloor \frac{T_{old}}{2} \rfloor, T_{min}),$$

where $T_{min}$ is a lower bound set to 2. This method ensures that when the transmission session transitions from CPU-limited to bandwidth-limited state, the frequency of ACKs is updated accordingly to avoid impairing throughput.

## 5 Evaluation

In this section, we evaluate HTTP/3 with our LiteQUIC in both data transmission and DASH-based streaming sessions.

**Implementation:** We implement the ACK merge and adaptive ACK frequency mechanism in nginx-quic. We replace `recvmsg()` with `recvmmsg()`. After reading packets from the kernel, packets from different connections are separated and the ACK frames from the same connection are merged for further processing. When merging ACK frames, we only merge their ACK ranges. Occasionally, ACK frames carry processing delays of client and ECN counts, which are handled as the vanilla QUIC. For the adaptive ACK frequency mechanism, both the server-side nginx-quic and the client-side nghttp2 have undergone modifications. We add the new transmission parameter, min_ACK-delay, to be exchanged during connection establishment to ensure mutual support. The server sends the `ACK-FREQUENCY` frame only if both sides support this feature. The nghttp2 client is added a module to handle `ACK-FREQUENCY` frame, extracting the updated value from the server to adjust the frequency of sending ACK frames.

### 5.1 Data Transmission

We first evaluate the performance of LiteQUIC in terms of data transmission, with the experimental setup identical to that in Section 3. We compare it against picoquic, ngtcp2 [49], lsquic [42], quic-go [53], xquic [1], msquic [45], quicly [26] and mvfst [22]. Most of them have participated in the interoperability tests [54] of QUIC and have high stars on GitHub. We evaluate two versions of LiteQUIC in this experiment. LiteQUIC+ means we only incorporate ACK merge, and LiteQUIC++ additionally adopts adaptive ACK frequency. We also test HTTP/3 with optimizations. HTTP/3 is the vanilla version, +GSO means we activate GSO, and +PicoTLS means have both GSO and PicoTLS in HTTP/3.

We first set the ACK threshold to 2 and 10, which is currently adopted by most QUIC implementations [18], and simultaneously initiate 10 client requests for a 1GB file, recording their total throughput. The results are shown in Figure 7, and we see that when the ACK threshold is 2, the throughput of HTTP/3 without any optimization is 1.79Gbps. Even after adding GSO and PicoTLS optimizations, the throughput is only 2.28Gbps, which is only a 27.3% increase. This is because, as mentioned earlier, the server can only release space for two packets in the sending window after receiving an ACK, and GSO cannot effectively combine multiple packets for sending, thus does not work as expected. When ACK merge is used, in CPU-limited states, the server not only reduces the overhead of ACK processing but also releases more sending window, which allows GSO work effectively, leading to a throughput of 5.20Gbps in LiteQUIC+. This is 2.28× of HTTP/3 with GSO and PicoTLS, and a 20.3% improvement over picoquic and quicly. When additionally

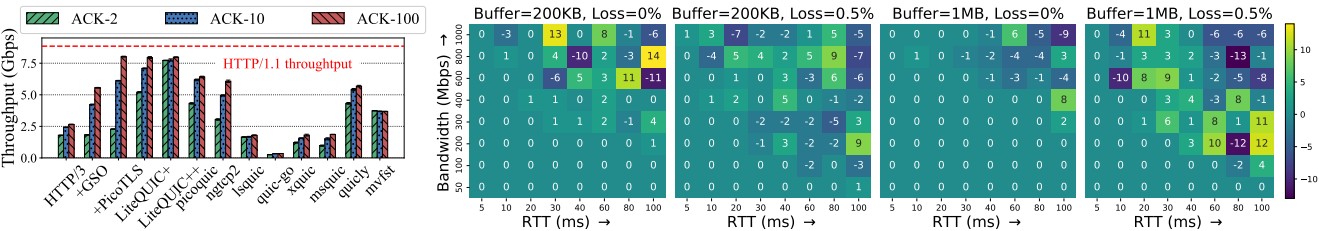

**Figure 7: Left:** Throughput of different QUIC implementations. **Right:** The throughput gain of LiteQUIC++ compared to itself with the ACK threshold is fixed to 2.

using the adaptive ACK frequency mechanism, the throughput of LiteQUIC++ reaches 7.72Gbps, which is 87.2% of HTTP/1.1, a 48.4% improvement over LiteQUIC+, and a 78.7% improvement over picoquic and quicly.

When the ACK threshold is set to 10, which is the by default configuration of Chrome browsers, HTTP/3 achieves a throughput of 2.43Gbps. At this point, the bottleneck lies in packet sending and data encryption. Significant improvements can be observed when utilizing GSO and PicoTLS, resulting in a throughput of 6.11Gbps, marking a 2.51× enhancement compared to without them. The performance of HTTP/3 closely approaches that of picoquic, which is the best among QUIC implementations at 6.17Gbps. LiteQUIC+ demonstrates a throughput of 7.09Gbps, surpassing HTTP/3 with GSO and PicoTLS by 16%. LiteQUIC++ achieves the highest throughput at 7.75Gbps, a 9% improvement over LiteQUIC+.

When the ACK threshold is set to 100, the overhead of ACK processing is supposed to be significantly reduced. However, with this optimization, the throughput of HTTP/3 remains as low as 2.65Gbps, showing only a 9% increase compared to when the ACK threshold is set to 10. This suggests that the bottleneck at this stage is not primarily due to ACK processing. With the integration of GSO and PicoTLS, the proportion of overhead caused by ACK processing becomes more prominent. Therefore, reducing the ACK frequency leads to substantial benefits. In this scenario, a throughput of 8.02Gbps is achieved, marking a 31% improvement over the performance with an ACK threshold of 10. LiteQUIC+ and LiteQUIC++ achieve throughputs of 7.94Gbps and 7.97Gbps, respectively, closely rivaling HTTP/3 with GSO and PicoTLS, and both outperform other QUIC implementations. However, as mentioned in Section 3.1, simply reducing the ACK frequency can also lead to performance degradation in poor network conditions. LiteQUIC++ only reduces the ACK frequency when CPU is limited, thus not impacting throughput in bandwidth-limited scenarios. Analogous to Section 3.1, we also report the throughput gains of LiteQUIC++ compared to itself but always setting ACK threshold to 2 under different network conditions. The results are shown in the right of Figure 7. The throughput of LiteQUIC++ is similar to setting the ACK threshold to 2, without the performance degradation that occurs when the ACK threshold is set to 100.

## 5.2 Dash Video Streaming

We also assess LiteQUIC for DASH-based streaming sessions compared to HTTP/1.1 and HTTP/2. Both HTTP/1.1 and HTTP/2 use TLS/TCP, and their performance is almost similar, so we chose HTTP/1.1 as the representative. We deploy our DASH client in the Chrome browser, and the server based on nginx-quic. Since QUIC implemented in Chrome does not support the `ACK-FREQUENCY` frame, only ACK merge is effective in LiteQUIC.

**Experimental setup:** We build a DASH-based video streaming system based on HTTP/3 protocol. We use a 5-min video Red Bull[1], which is encoded into {1, 5, 10, 20, 30, 40, 50}Mbps using FFmpeg, and is further split into 10s segments using GPAC MP4Box [24]. We use *dash.js*, the most widely used browser-based DASH player, as the client. We carry out the experiments with different ABR algorithms, including Dynamic [61], BOLA [62], Throughput rule, L2A [35], and LoLP [40]. Throughput rule is a naive method that chooses the highest bitrate lower than the bandwidth measured most recently.

**Metrics:** We use two metrics, average bitrate (AB) and rebuffering time (RT), to evaluate the QoE.

- *Average bitrate*: The average bitrate (from 1Mbps to 50Mbps) of all played segments. The higher the AB, the higher the QoE.
- *Rebuffering time*: The total time of rebuffering during playback. The higher the RT, the lower the QoE.

We gather QoE metrics across various number of clients, various ABR algorithms, and in poor network conditions. Figure 8 shows the average bitrates, while Figure 9 displays the rebuffering time.

*5.2.1 Various number of clients:* Using the Dynamic algorithm with 20 clients, even without any optimizations, HTTP/3 can adequately meet video transmission requirements. However, as the number of clients increases to 40, CPU resources become insufficient, resulting in a 23% decrease in AB and introducing a 0.96-second RT. As the number of clients increases to 80, both +GSO and +PicoTLS fail to meet bandwidth requirements. Nonetheless, LiteQUIC still maintains a satisfactory watching experience, achieving an AB of 47.68Mbps and an RT of only 0.17 seconds, marking a 19% improvement in AB and an 83% reduction in RT compared to +PicoTLS. Upon reaching 100 clients, LiteQUIC experiences a decline in QoE, yet it still outperforms +PicoTLS. Its AB reaches 41.24Mbps, which is 88% of HTTP/1.1, surpassing +GSO and +PicoTLS by 48% and 20%, respectively. Moreover, LiteQUIC maintains a low RT of 0.22 seconds, representing a 97%, 96%, and 93% reduction compared to HTTP/3, +GSO, and +PicoTLS, respectively. Notably, although LiteQUIC falls behind HTTP/1.1 in AB, it achieves lower RT, with average rebuffering times of 0.13 seconds for 20 to 100 clients, compared to 0.23 seconds of HTTP/1.1, a reduction of 43%.

[1]Available at https://dash.akamaized.net/akamai/customerconference2013/redbull/

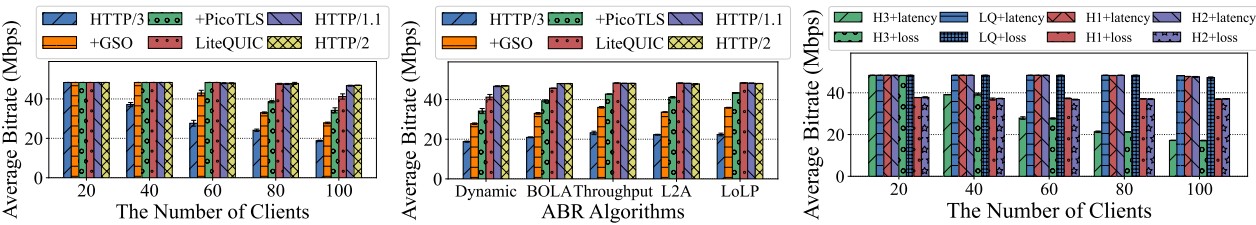

**Figure 8: The average bitrate (AB) of DASH-based streaming sessions.** +GSO represents the activation of GSO optimization for HTTP/3, and +PicoTLS represents the further utilization of PicoTLS on top of enabling GSO.

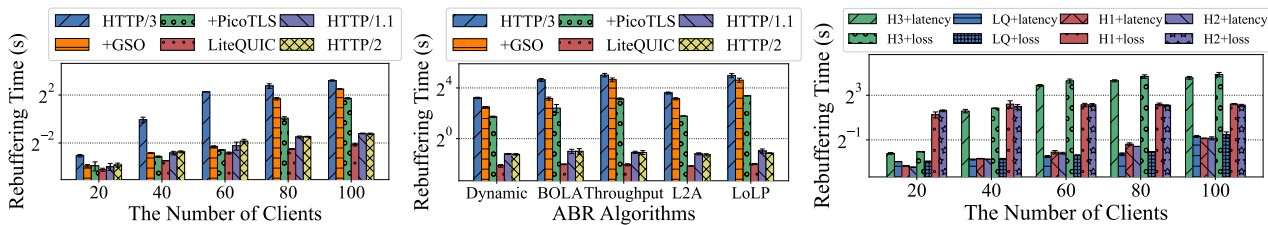

**Figure 9: The rebuffering time (RT) of DASH-based streaming sessions.** +GSO and +PicoTLS have the same meaning as in Figure 8.

This is primarily attributed to the latency introduced during the startup phase, whereas the 0-RTT handshake of QUIC facilitates rapid connection establishment, resulting in low startup latency.

*5.2.2   Various ABR algorithms:*  At 100 clients, with different ABR algorithms, Dynamic achieves the lowest RT, while the highest RT is observed in HTTP/3 without any optimizations at 9.27 seconds. Throughput rule and LoLP receive the similarly high RTs, which are 32.37 seconds and 31.63 seconds, respectively, leading to inferior QoE. Meanwhile, LiteQUIC continues to maintain the lowest average RT of 0.23 seconds, a reduction of 50% compared to HTTP/1.1 and 96% compared to +PicoTLS. Regarding AB, the performance trend of HTTP/3 with different optimizations is consistent. LiteQUIC consistently outperforms +PicoTLS across all ABR algorithms, with up to a 20% improvement observed in Dynamic. Specifically, in terms of Throughput rule, L2A, and LoLP, LiteQUIC achieves AB values of over 48.30Mbps, marking increases of 17% compared to Dynamic. Overall, when throughput is insufficient, Dynamic performs better by adopting a more conservative strategy, ensuring a lower RT. On the other hand, Throughput rule and LoLP introduce excessively high RTs, severely degrading the user viewing experience. Conversely, when throughput just satisfies requirements, the conservative strategy of Dynamic leads to lower AB compared to other ABR algorithms with LiteQUIC. For example, L2A with LiteQUIC reaches both high AB and low RT.

*5.2.3   Poor network conditions:*  We introduce a latency of 20ms and packet loss of 0.5% to compare the performance in a weak network environment. Using L2A as the ABR algorithm due to its high bandwidth utilization and low rebuffering time. When the latency is introduced, there is no significant change in performance. However, with the introduction of packet loss, a notable deterioration in the performance of HTTP/1.1 is observed. Due to

the elimination of head-of-line blocking and wider acknowledgment scope compared to TCP, QUIC exhibits superior performance in high packet loss scenarios. With 20 clients, even unoptimized HTTP/3 outperforms HTTP/1.1. At this point, it achieves 1.28× the AB of HTTP/1.1 and reduces RT by 89.9%. As the number of clients exceeds 60, HTTP/3 enters a CPU-limited state, unable to match available bandwidth due to its overhead, resulting in performance dropping below HTTP/1.1. However, by reducing CPU overhead, LiteQUIC maintains a high QoE even with 100 clients and performs well in weak network environments. At this stage, it achieves 1.28× the AB of HTTP/1.1 and lowers RT by 85.2%. In summary, HTTP/1.1 has low CPU overhead but suffers in high packet loss scenarios. The vanilla QUIC achieves superior performance in poor network conditions but its performance quickly degrades as more clients are connected. With our optimizations on reducing CPU overhead, LiteQUIC achieves high QoE no matter in a poor network scenario or with a large number of clients connected.

## 6   Conclusion

In this work, we identify three major sources of CPU overhead in QUIC: ACK processing, packet sending, and data encryption. By reducing the ACK frequency, activating UDP GSO, and incorporating a high-performance encryption library, the CPU overhead of QUIC could be effectively reduced. However, directly reducing the ACK frequency leads to low throughput in poor network environments and GSO does now work well with high ACK frequency. Therefore, we propose LiteQUIC, a lightweight QUIC for complex network environments that has the server-side ACK merging and the client-side adaptive ACK frequency mechanism. We test LiteQUIC in both data transmission and DASH-based streaming sessions. The results show that LiteQUIC achieves high throughput and QoE, regardless of whether the server CPU is heavily crammed due to an increasing number of connected clients or the network environment is poor.

## Acknowledgments

This work is supported in part by the National Natural Science Foundation of China (No.62102229), the Natural Science Foundation of Shandong Province, China (No.ZR2022ZD02), and the Qingdao Municipal Science and Technology Bureau (No.23-1-2-qljh-8-gx).

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
