# OpenReview forum: "LiteQUIC: Improving QoE of Video Streams by Reducing CPU Overhead of QUIC"
_acmmm.org/ACMMM/2024/Conference — MM2024 Oral_

### Official Review · Reviewer_9fwK · 2024-05-10

**Rating:** 5
**Confidence:** 4

**Summary:**

This paper presents LiteQUIC, an improved version of QUIC and its evaluation form HTTP/3 implementation against HTTP 1.1; and its performance for DASH. It is a ten pages paper that clearly presents the problems of QUIC and some solutions that are well presented and explained. They also make good comparison with other similar frameworks.

**Strengths:**

The clarity of presentation is the best value of this paper. They have also reference good and interesting papers (they have treated a traditional and classical problem).
They present a significative number of results they explain good the advantages and disadvantages of their results.
The paper is technically sound.

**Limitations:**

Fix some grammar errors.
Expressions like "5) Etc.: Any other overhead." are not appropriated for a scientific paper. I would remove it.
What about the performance of LiteQUIC if you will allow more or less than an additional 0.5% random packet loss ratio? Please discuss.
I think you could improve the readability of your paper if you improve the presentations and discussion of the results and comparisons: It is hard to read that vast number of numerical results without any convincing and technical opinion explaining why the results are the ones you obtained. Conclusions must be improved.

**Suitability:**

3

---

### Official Review · Reviewer_b7mP · 2024-05-24

**Rating:** 2
**Confidence:** 4

**Summary:**

This paper analyzes the computational overhead issues of QUIC and identifies the performance bottlenecks in ACK processing, packet transmission, and data encryption. Subsequently, the article proposes LiteQUIC, which addresses these issues by leveraging UDP generic segmentation offload, integrating the high-performance encryption library PicoTLS, and adapting ACK frequency to reduce QUIC's computational demands. Finally, the paper conducts experimental evaluations of LiteQUIC in file transfer and DASH-based video streaming scenarios. The evaluation results demonstrate that LiteQUIC effectively reduces CPU consumption and improves file transfer speed in multi-user concurrent access scenarios. In video streaming scenarios, LiteQUIC increases the average bitrate and reduces rebuffer times.

**Strengths:**

* The paper is easy to follow.
* LiteQUIC is implemented upon different versions of QUIC.
* LiteQUIC provides an improvement in QoE in some scenarios.

**Limitations:**

* Improper handling of ACK delay field.
* Lack of research on state-of-the-art techniques.
* Inadequate validation in large-scale real-world environments.

**Suitability:**

2

---

### Official Review · Reviewer_Th9R · 2024-05-24

**Rating:** 5
**Confidence:** 3

**Summary:**

Triggered by observation of bottleneck behaviors when using the popular streaming protocol QUIC, the authors propose and validate a set of measures to reduce such suboptimal throughput behavior. First, they reduce the acknowledgment (ACK) intensity of QUIC, which has a significant effect but in adverse network conditions that are emulated through realistic delay and loss values. Furthermore, the effects of introducing merging of ACKs and adapting their frequency in correspondingly configurable clients is studied. Second, they investigate the introduction of Generic Segmentation Offload (GSO), which shows to be the more efficient, lower the ACK intensity becomes. Third, they introduced a lightweight encryption solution (PicoTLS). These innovative elements form ingredients of the proposed LiteQIUC protocol, whose various versions are benchmarked against other available versions.

**Strengths:**

The paper starts off from a relevant problem that can be observed in practice, namely suboptimal streaming throughput performance that is due to end-node processing rather than to network problems.

The approach is systematic, well-designed and well-documented. It also becomes clear how the different proposed additions to QUIC have to be combined or developed further in order not to end up in some performance trap again (e.g. due to bad network conditions).

Figure 2 (left) is providing valuable insights into the distributions of various delay contributions for the various configurations.

Most of the measurement results of average bitrate and rebuffering times are shown together with confidence intervals (which is all but self-understood, but needed to establish credibility in the results).

**Limitations:**

There are some issues regarding
- Credibility: Figures 1 and 3 would do good from confidence intervals.
- Readability: The numbers in figures 2 (right) and 7 (right) are hardly legible in a print-out -- it would be helpful if they could be enlarged

Minor corrections:
- Line 111: impair => impairs
- Lines 340-341: Considering the subsequent formula for TPGain, the description should refer to ACK-2 as compared to ACK-100 (as this is in the denominator)

**Suitability:**

2

---

### Official Review · Reviewer_Qenc · 2024-05-24

**Rating:** 4
**Confidence:** 3

**Summary:**

This paper examines QUIC based on UDP as the underlying protocol of HTTP/3, which is important for delivering video data nowadays, mainly from a server's perspective. Through experiments, the authors find that QUIC incurs high computational overhead. When CPU resources become the constraint, HTTP/3 exhibits lower throughput than HTTP/1.1. They analyze the performance bottlenecks as ACK processing overhead, packet sending overhead, and data encryption overhead. Then, they try out several mitigation solutions, including reducing ACK frequency, enabling UDP GSO, and incorporating PicoTLS, which are shown to effectively reduce CPU overhead in stable network conditions. However, they find that simply reducing ACK frequency also reduces throughput under poor network conditions. To further address the problem, they develop LiteQUIC with two mechanisms for alleviating ACK processing overhead: ACK merge and adaptive ACK frequency, in addition to using GSO and PicoTLS. They implement and evaluate LiteQUIC in both file download and DASH video streaming scenarios, demonstrating significant performance improvements.

**Strengths:**

This paper provides a timely study on QUIC, as QUIC is trending and set to replace the TCP protocol stack. It provides clear descriptions of the experimental setup and detailed evaluations including preliminary measurements and LiteQUIC evaluations. The preliminary analysis of the CPU overhead well motivates the design of LiteQUIC. The proposed solutions are feasible with no requirements for client modifications, enhancing the potential for real-world deployment.

**Limitations:**

Although the paper already shows that the original QUIC and HTTP/3 perform worse than HTTP/1.1, it is not explained why HTTP/2 was not included in the comparisons given that HTTP/2 is more prevalent and well-maintained than HTTP/1.1. Including HTTP/2 would provide a more comprehensive analysis of QUIC and evaluations of the improvements brought by LiteQUIC.

The paper would benefit from a deeper analysis of the CPU overhead. Currently, it is just stated that the three parts are major sources of CPU overhead with an overhead breakdown. Some additional details regarding how these components are measured (e.g., specific function calls) would be helpful and make the analysis more convincing.

The experiments use a maximum bitrate of 50 Mbps, which may not meet the latest high-quality streaming requirements for 4K/8K resolution or 360-degree videos.

The paper relies on a 10Gbps link or tc-emulated experiments (please clarify if you have other configurations for later experiments) to simulate network conditions. It would be great if you could add results from real-world environments. They would provide stronger validation of the proposed solutions.

Typos: Sec 3.1, "are" is missed in "what are the impacts of reducing ACK frequency?".

**Suitability:**

3

---

### Meta-Review · Area_Chair_xvqF · 2024-07-02

**Recommendation:** Accept (Oral)
**Confidence:** 5

**Metareview:**

The paper presents some shortcoming of QUIC and some methods to address them. This has some practical implications for a protocol that is deployed at scale. All the reviewers agree on the benefits of the solution and on the technical merits of the paper, and therefore recommend some form of acceptance (borderline, weak or plain accept).